# Study on Driving Factors and Spatial Effects of Environmental Pollution in the Pearl River-Xijiang River Economic Belt, China

**DOI:** 10.3390/ijerph19116833

**Published:** 2022-06-02

**Authors:** Yutian Liang, Jiaxi Zhang, Kan Zhou

**Affiliations:** 1School of Geography and Planning, China Regional Coordinated Development and Rural Construction Institute, Sun Yat-sen University, Guangzhou 510275, China; lytian@mail.sysu.edu.cn (Y.L.); zhangjx235@mail2.sysu.edu.cn (J.Z.); 2Key Laboratory of Regional Sustainable Development Modeling, Institute of Geographic Sciences and Natural Resources Research, Chinese Academy of Sciences, Beijing 100101, China

**Keywords:** environmental pollution, spatial effects, spatial error model, the Pearl River-Xijiang River Economic Belt (PRXREB)

## Abstract

As a typical basin area in China, the Pearl River-Xijiang River Economic Belt (PRXREB) faces multiple types of environmental problems caused by the different development conditions of basins. To identify the situations of environmental pollution in the PRXREB, this paper constructed the Environment Pollution Composite Index (EPCI) by using four environmental pollutant emission indicators based on the entropy weight method, and explored the spatial effects and driving factors of environmental pollution by using the Spatial Error Model (SEM). The results showed that: (1) EPCI of the PRXREB decreased significantly from 2012 to 2016, and the spatial patterns were relatively stable. Wherein, the midstream and downstream were always the critical areas of environmental pollution. (2) Spatial spillover effects were significant in the PRXREB, which revealed that the local environmental pollution degree was affected by adjacent areas. (3) Industrial structure, infrastructure construction, and regulatory measures were the main driving factors of environmental pollution in the PRXREB. (4) To balance economic development and environmental protection in basin areas, environmental regulations such as environmental access, pollution payment, and cross-border early warning should be jointly established.

## 1. Introduction

Since the reform and opening up in 1978, the Pearl River-Xijiang River Economic Belt (the PRXREB) had entered the development stage of rapid industrialization and urbanization. The spatial expansion of high speed, high density, and high intensity would inevitably lead to problems of high energy consumption, high emission, and high pollution [1]. The PRXREB connects the developed areas in the eastern part with the less developed areas in the western part of China. As the economic frontier of China, the Pearl River Delta region, which is downstream of the PRXREB, had developed rapidly. At the same time, the structure of pollution-intensive and energy-intensive industries led to the increasingly prominent contradiction between economic growth and environmental pollution [2]. The upstream and midstream of the PRXREB, which are located in Guangxi Autonomous Region also had problems with industrial pollutant emissions [3], which not only destroyed their ecological environment foundation of sustainable development [4,5] but also directly threatened the ecological security and environment quality of the downstream. In exploring a new model for cross-provincial basin ecological construction, analyzing the spatial-temporal change process, main driving factors, and spatial effects of environmental pollution in the PRXREB could provide a scientific reference for policy formulation. 

So far, existing studies on environmental pollution mainly focused on the following aspects. (1) Spatial–temporal variation and spillover effects of environmental pollution [6,7], which were mainly based on single pollutant emission or multi pollutants emissions. Many studies showed that the agglomeration of population and industry led to the spatial concentration of pollutant emissions, such as PM2.5 [8,9] and heavy metal pollutants emissions [10] in urbanized regions [11,12] and industrial areas [13,14,15]. Moreover, research demonstrated that environmental pollution had significant spillover effects among regions based on spatial econometric models. It caused not only contiguous regional pollution [16,17] but also had adverse impacts on public health [18,19]. So, researchers proposed that regional joint prevention and control should be taken into consideration urgently [20,21,22]. (2) The driving factors of environmental pollution. Since the Kuznets inverted U-curve between economic development level and environmental pollution was proposed, environmental pollutant emissions had attracted scholars’ attention as a by-product of economic growth [23,24]. Then, the consideration of driving factors gradually expanded to other aspects of the economy, such as industrial structure, international trade [25], and foreign direct investment [26]. Some studies analyzed the spatial diffusion of environmental pollution by carefully considering spatial patterns and driving factors to explain local pollution. Many pieces of research proved that environmental pollution had significant spatial aggregation and diffusion effects due to geospatial attributes and regional economic association [27,28]. However, policy formulation required researchers to downscale and carry out micro research on the spatial–temporal evolution and driving forces of environmental pollution.

This study focused on environmental pollution in the PRXREB, which is one of China’s typical basin areas, and analyzed its spatial–temporal variation at the basin scale and the county scale. Then, the driving factors and spatial effects were explored at the county scale. Specifically, we picked four pollutants emission indicators, including chemical oxygen demand (COD), ammonia nitrogen (NH_3_-N), sulfur dioxide (SO_2_), and nitrogen oxide (NO_x_), to construct a comprehensive index of environmental pollution (EPCI), which could reveal the environmental pollution degree. Additionally, the driving factors and spatial effects of environmental pollution were quantitatively decomposed with spatial econometric models. Based on the above steps, this research could contribute to understanding the characteristics of environmental pollution in basin areas, and provide scientific reference for policy formulation, which could enhance environmental management and control capacity from the source in basin areas.

## 2. Materials and Methods

### 2.1. Case Overview

The PRXREB includes 11 cities, which are Guangzhou, Foshan, Zhaoqing, Yunfu in Guangdong province, and Nanning, Liuzhou, Wuzhou, Guigang, Baise, Laibin, and Chongzuo in Guangxi Autonomous Region, with altogether 83 county administrative units. The total area of this region is 165,000 km^2^ (Figure 1). In 2013, the population and the GDP of the PRXREB were 52.28 million and 3.87 trillion yuan, relatively. The pollutant emissions and socio-economic database in the PRXREB of 2012–2016 were established in county administrative units.

### 2.2. Data Sources

The primary data sources were as follows: the administrative boundary was obtained from the website of the National Basic Geographic Information System. The pollutant emissions and socio-economic data were obtained from China Environmental Statistical Yearbook, China Environmental Yearbook, China City Statistical Yearbook, China Regional Economic Statistical Yearbook, and provincial statistical yearbook.

### 2.3. Environmental Pollution Composite Index (EPCI)

The computational procedures of EPCI based on the entropy weight method were as follows: firstly, the extremum method was adopted to standardize the pollutant emission index *r_ij_* of counties, in which *r_ij_* indicated the value of pollutant emission of *j*-th pollutant in *i*-th county, then the dimensionless attribute value *r*′*_ij_* was obtained. Secondly, the entropy weight was calculated as *E_j_*:(1)$$E_{j}=-k\sum R_{ij}*lnR_{ij}$$

In Formula (1), *R_ij_* = *r*′*_ij_/*Σ*r*′*_ij_*. The number of the counties in the PRXREB was *n*, the number of the pollutant emissions indicators was *m*, made *k* = 1/*ln*(*n*). Then, the weight of each index was calculated as *W_j_* = (1 − *E_j_*)/(*m* − Σ*E_j_*). Finally, the Environmental Pollution Composite Index (EPCI) was obtained:(2)$$EPCI_{i}=\sum W_{j}*R_{ij}$$

In Formula (2), *EPCI_i_* could reflect environmental pollution degree in a county. The greater the value of EPCI, the higher the pollutant emission intensity, and the greater the pressure of pollutant emission faced by a county.

### 2.4. Models

#### 2.4.1. Model Construction and Variable Selection

Using existing studies for reference and considering the availability of county-level data, this research picked some variables to reflect several aspects of social and economic development, which could potentially lead to environmental pollution. Furthermore, given the basin area, possible spatial spillover effects of water and air pollutants emissions were considered. The relationship and mechanism of variables, aspects of development, and environmental pollution were shown in Figure 2. 

The initial general linear regression model was set as follows:lnEPCI = *α* + *β*lnPGDP + *η*lnRS + *γ*lnIS + *ζ*lnFAI + *θ*lnFED + *ε*(3)

In Formula (3), EPCI was the Environmental Pollution Composite Index, reflecting the situations of environmental pollution. PGDP was per capital GDP (yuan/person). Wherein, the deflator was calculated by taking 2012 as the base period to eliminate the impact of inflation on GDP. RS was the total retail sales of consumer goods. The above two variables could reflect the economic development levels. IS was the proportion of the secondary industry’s added value (%), reflecting the industrial structures. FAI referred to the total investment in fixed assets (100 million yuan), reflecting the investments in infrastructure construction. FED represented the decentralization degree of local fiscal expenditure, revealing local regulatory measures. It was the ratio of local per capital general fiscal budget expenditure and provincial per capital general fiscal budget expenditure. *ε* was the error term.

#### 2.4.2. Spatial Econometric Models

Spatial econometric models were used to explore the driving factors of environmental pollution. Specifically, the selection of models started from the ordinary least squares (OLS), which was strictly according to relative parameters proven in known research [29,30].

When the dependent variable had the spatial lag effect, the spatial lag term of the dependent variable (*W_Y_*) was added to the general linear regression model shown in Formula (3), and then the spatial lag model (SLM) was transformed as follows:(4)$$Y=\rho W_{Y}+X\beta +\varepsilon$$

When the error term was spatially dependent, the error term of the model was spatially autocorrelated. Added the spatial autocorrelation error term into the general linear regression model, and converted the OLS into the spatial error model (SEM) as follows:(5)Y=Xβ+μ

In Formula (5), the generating process of the error term (*μ*) was:(6)$$\mu =\lambda W_{\mu }+\varepsilon ,\;\;\varepsilon \sim N\left(0,\sigma^{2}I_{n}\right)$$

In Formulas (4)–(6), *W* was a spatial weight matrix, which was constructed based on the queen adjacency. *X* was a data matrix based on the sample size and the number of independent variables. *β* was the corresponding coefficient matrix. *ρ* was the spatial autoregressive coefficient. *λ* was the spatial autocorrelation coefficient between regression residuals. *ε* was the random error vector. The maximum likelihood (ML) method was used to estimate the parameters of each model.

## 3. Results

### 3.1. Spatial Pattern of Environmental Pollution

EPCI in the PRXREB declined significantly from 2012 to 2016. The mean values of EPCI in 2012 and 2016 were 0.137 and 0.095, relatively, indicating that the environmental pollution degree decreased by 30.6% during this period. At the same time, the spatial patterns of environmental pollution were stable from 2012 to 2016.

#### 3.1.1. Spatial Pattern of Environmental Pollution at the Basin Scale

At the basin scale, the mean estimated values of EPCI of the upstream, midstream, and downstream were 0.093, 0.136, and 0.242 in 2012, showing that environmental pollution degree increased from upstream to downstream. There was a trend of similarities between 2012 and 2016. The upstream includes the Youjiang River basin, Zuojiang River basin, and Yujiang river basin, which had the advantage of ecological resources but were less developed than the midstream and downstream in the economy. The mean values of EPCI of the upstream were lower than the overall average in both 2012 and 2016. In 2016, it was only 67.67% of the average level of the PRXREB. The midstream includes the Liujiang River basin, Qianxun River basin, Xijiang River basin, and Red River basin, which was more developed than the upstream but less developed than the downstream. From Figure 3, we could see that the midstream was lower than the PRXREB in average values of EPCI in 2012 and 2016. In contrast, the Red River basin therein had higher mean values than the PRXREB. Concretely, the mean values of the Red River basin reached 0.171 and 0.145 in 2012 and 2016, which were 1.23 and 1.46 higher than the average degree of the PRXREB. It was likely that the characteristics of EPCI echoed the reality that some industrial towns in the midstream stimulated local economic development and caused environmental pollution at the same time. Downstream, the Pearl River Delta basin is one of China’s most significant metropolitan areas. Due to the high development level, EPCI values of the downstream came up to 0.242 in 2012 and 0.195 in 2016, 1.74 and 1.96 higher than the average level of the PRXREB. The above results showed that the midstream and downstream were taking higher pressure of environmental pollution than the upstream, which meant these two basins were the key areas of environmental prevention measures and control.

#### 3.1.2. Spatial Pattern of Environmental Pollution at the County Scale

Firstly, using the grading method of Jenks natural breaks, the values of EPCI were divided into five grades at the county level as shown in Figure 4. Overall, the number of sub-high-level counties and medium-level counties reduced significantly, revealing that the environmental pollution degree in the PRXREB tended to decrease from 2012 to 2016. Specifically, the number of sub-high-level counties dropped from 11 to 4. Moreover, by comparing the spatial pattern of EPCI in 2016 with it in 2012, we found that the environmental pollution always showed a core–edge structure with a high-level grade of EPCI at the core and the levels of grades decreasing outwardly.

Secondly, the global Moran’s I was used to measure spatial autocorrelation of environmental pollution in the PRXREB. The values of the global Moran’s I were calculated based on both feature’s locations and values simultaneously, and then used to evaluate whether the pattern expressed was clustered, dispersed, or random. If the values in the dataset tended to cluster spatially, Moran’s index would be positive. The global spatial autocorrelation analysis at the county level showed that the global Moran’s I of EPCI were 0.3817 and 0.3385 in 2012 and 2016. These values passed the significance test (*p* < 0.01), revealing a significant spatial autocorrelation of environmental pollution degree among counties.

Thirdly, the Getis-ord G* index was calculated to explore the agglomeration characteristics of environmental pollution in the PRXREB. The results included four types, “Hot,” Sub-hot,” “Sub-cold,” and “Cold.” If a county was displayed as “Hot” or “Sub-hot,” it meant this county had a higher value of EPCI than average, in which “Hot” was better higher than “Sub-hot.” Accordingly, if a county was displayed as “Cold” or “Sub-cold,” it indicated this county had a lower value of EPCI than average, in which “Cold” was lower than “Sub-cold.” For a better comparison of spatial patterns in 2012 and 2016, the types of changes were exhibited in Figure 5. What could be clearly seen in this figure is that many counties turned into lower agglomeration types from 2012 to 2016. Explicitly speaking, six counties of the “Hot” type in 2012 converted into the “Sub-hot” type in 2016, and 16 counties of the “Sub-hot” type in 2012 converted into the “Cold” type in 2016. 

Furthermore, we could also see a spatial pattern of core–edge structure in Figure 5. The “Hot” spots and “Sub-hot” spots of environmental pollution degree were mainly concentrated in industrial towns and urbanized regions such as Laibin city, Liuzhou city, and Nanning city in the midstream and Guangzhou city, Foshan city in the downstream. These regions usually had substantial degrees of industrial agglomeration or high levels of economic development, revealing industrialization and urbanization’s effects on environmental pollution. At the same time, the “Cold” and “Sub-cold” spots were at the edge of industrialized towns and urbanized regions. 

In summary, the measures of pollution control from the source should be more contra posed the core areas, which are located in industrial towns and urbanized regions. Additionally, measures should be focused on the guidance and promotion of cleaner production methods. At the same time, attention also should be paid to controlling the discharge of domestic pollution caused by the high density of the population. Moreover, the pollution diffusion from the midstream, downstream to upstream should be prevented. Therefore, a stricter negative list of environmental access should be implemented upstream. In addition, it was worth exploring if there was an apparent spatial spillover effect between core areas and edge areas. 

### 3.2. Driving Factors of Environmental Pollution

#### 3.2.1. Model Checking and Parameter Estimation

Considering the difference in environmental pollution degrees and development stages among basins and counties, the driving factors and spatial effects were analyzed at both the whole scale and local scale to show more detailed results. At the local scale, the PRXREB was divided into the Guangxi section and the Guangdong section by provincial boundary. Wherein, the Guangxi section represented the upstream and midstream, while the Guangdong section represented the downstream. 

The selection of models was processed according to existing research [29]. First, the ordinary least squares (OLS) regression model was used to test the spatial effects of environmental pollution in the PRXREB. Secondly, the applicability of econometric models was clarified with the significance of critical parameters. Then the best-fit model was selected for further analysis. 

The diagnostic results of the OLS were as follows (Table 1): Moran’s I (error) index of both the PRXREB and the Guangxi section had passed the significance test in 2012 and 2016. LM-lag and LM-Error statistics of these two areas in 2012 were also significant. In 2016, LM-lag and LM-error were significant in the Guangxi section, and LM-error was more significant than LM-lag. As for the PRXREB, LM-Error was significant, while LM-lag was not significant. In the Guangdong section, these statistics magnitudes were only significant in 2012. Further, the maximum likelihood estimation results showed that Lagrange multiplier LM-error and Robust Lagrange multiplier Robust LM (Error) of the SEM were higher than these parameters of the SLM. Meanwhile, the AIC and SC values of the SEM were smaller, and R^2^ and log-L values were higher than the SLM’s (Table 2, Table 3 and Table 4), in which the minimum value of R^2^ was 0.5036, indicating that the selected variables had strong correlations with environmental pollution [31]. 

The above results indicated that the SEM had better fitness than other models. Based on this, the SEM was introduced to analyze the driving factors of environmental pollution in the PRXREB. At the same time, the OLS and the SLM were also operated to compare and ensure accuracy.

From the testing results, it could be noted that there was spatial dependence of environmental pollution degrees in the PRXREB in 2012. As of 2016, spatial dependence mainly existed in the Guangxi section. The spatial dependence revealed that the environmental pollution degree had positive spillover effects on adjacent regions. That meant the change in pollution degree in a county would also lead to the same changes in nearby counties. The reasons for the effects may be as follows. On the one hand, neighboring counties were usually in the same regional economic system, sharing common labors and markets, and then developed similar industrial structures. If local pollution-intensive industries were difficult to clean up, the pollution in neighboring regions would also be in a bad situation. On the other hand, pollution transfer to nearby areas was proven to be widespread. Due to the combined effects of increasing returns to scale, consumer preferences, and logistics costs, pollution-intensive industries were unlikely to make a large-scale spatial transfer even if they were to relocate.

#### 3.2.2. Driving Factors of Environmental Pollution

On the whole scale, the results (Table 2) showed that the environmental pollution degree in 2012 was mainly affected by infrastructure construction and industrial structure. To be specific, every 1% increase in SFA would enhance EPCI by 0.3549%. That indicated it was urgent to strengthen the construction of sewage treatment facilities and pipe networks in the PRXREB, so as to integrate the environmental carrying capacity and cope with the environmental pollution caused by the rapid urbanization process at present and in the future. As for industrial structure, every 1% increase in IS would enhance EPCI by 0.6560%. It revealed the significant influence of high-speed industrialization on environmental pollutants discharge in the PRXREB, reflecting the threat to the environment caused by the high proportion of pollution-intensive industries. By 2016, economic development and industrial structure became the main driving factors of environmental pollution in the PRXREB. On the one hand, the driving effect of PGDP was negative, and every 1% increase in PGDP would decrease EPCI by 0.5287%. It reflected that a high economic level could decrease local environmental pollution degree. On the other hand, IS displayed positively, and the driving effects of this variable increased from 0.6560% in 2012 to 0.7419% in 2016, indicating the imperious demand for the implementation of industrial transformation and upgrading. More attention should be paid to some high-energy-consumed and high-pollution industries, such as chemical raw materials and products, non-metallic mineral products, agricultural and sideline products processing, metal smelting, and electricity, heat production, and supply. In addition, pollutant discharge reduction targets and environmental benefits should be brought into the evaluation mechanism of production capacity and technology.

At the local scale, two sub-models were carried out for the Guangxi section (Table 3) and the Guangdong section (Table 4). 

As for the Guangxi section, the results were as follows. In 2012, environmental pollution degree was mainly affected by industrial structure and infrastructure construction, and both of them were positive. Specifically, every 1% increase in IS would enhance EPCI by 0.8583%. Every 1% increase in SFA would cause a corresponding increase of 0.7250% in EPCI. By 2016, economic development, industrial structure, and regulatory measures all had significant and positive effects on environmental pollution degree, wherein, every 1% increase in PGDP, IS, SC, and FED would enhance EPCI by 0.5676%, 0.9426%, 0.3839%, and 1.0125%, respectively. As a whole, the industrial structure and economic development were fundamental reasons for the emissions of environmental pollutants in the Guangxi section. Moreover, the driving effects of the industrial structure showed an increasing trend. At the same time, the states of regulatory measures changed from insignificant to significant, revealing that local governments seemed to have the incentive of lowering environmental standards to promote economic development, although it may lead to an increase in environmental pollution degree. 

In the Guangdong section, the effects of industrial structure and infrastructure construction were also significant and positive in 2012. Every 1% increase in IS and SFA could enhance EPCI by 0.5898% and 0.5754%, relatively. By 2016, the effects of these two factors were no longer significant, with the effect of regulatory measures beginning to emerge. Every 1% increase in FED would cause a 0.7008% increase in EPCI. That was to say, local governments in the Guangdong section also had the problem of lowering environmental pollution standards to keep the economy sustainable.

A comprehensive comparison of the driving factors at different scales showed that the main driving factors in the PRXREB were industrial structure, infrastructure construction, and regulatory measures, which were presupposed in Figure 2. Rapid industrialization, accelerating urban construction, and local government pursuing economic development instead of environmental protection led to environmental pollution. Since the Guangdong section started early and had the advantage of economic development and production technology over the Guangxi section, the effects of the above three factors in the Guangdong section were mainly revealed in 2012. By 2016, only regulatory measures were the main driving factors. In contrast, there was a strengthening trend of the effects of these three factors in the Guangxi section. The reasons could be speculated as follows. First, the Guangxi section was less developed than the Guangdong section and was urgently inclined to pursue rapid economic growth through industrialization, which inevitably led to environmental pollution. Second, considering the geographical proximity between the Guangdong section and the Guangxi section, the latter was likely to carry on industrial transfer from the frontier, which could also lead to environmental pollution. 

## 4. Discussion

This study focused on the PRXREB, which is one of China’s typical basin areas, analyzed its comprehensive pollution process of various anthropogenic pollutant emissions, expecting to explore the characteristics and driving factors of environmental pollution in basin areas.

It could be found that the downstream usually started the process of rapid urbanization and industrialization at first, and the development with time–space compression characteristics arose environmental pollution simultaneously. With the development of the economy and technologies, the industrial center began to transfer from the downstream to midstream, leading to a similar problem for the latter. At the same time, significant spatial spillover effects could be detected in this basin area. The increase in local pollution degree would aggravate the pollution degree in neighboring areas. As for this, the implementation of industrial transformation and upgrading is urgent. In the future, emission reduction targets and environmental benefits should be considered in evaluation systems of production capacity and production technology. Meanwhile, systematic and comprehensive treatments should be implemented in industrial structures and modes of production to alleviate environmental pollution from sources. In addition to that, green and low-carbon modes of production and life should also be advocated. Since upstream usually assumed an ecological function, it is necessary to avoid possible pollution transfer to these areas. In response, deeper environmental regulations, such as environmental access, pollution payment, and cross-border early warning, should be established in coordination.

It should be noted that this study still leaves much to be desired. Due to the limitation of county-level data acquisition, the continuity and timeliness of data are insufficient, and the emissions of other pollutants such as soil and solid are not included. In the future, an environment monitoring system could be established based on the data of the second national census of pollution sources, including total factors and a panel database of space and time. Then, more comprehensive studies of regional environmental pollution could be conducted in a long time series.

## 5. Conclusions

Firstly, environmental pollution degrees in the PRXREB decreased significantly from 2012 to 2016, since the mean values of EPCI inclined from 0.137 to 0.095, with a decrease of 30.6%. At the same time, the spatial patterns of environmental pollution kept relatively stable during the five years. At the basin scale, environmental pollution degrees showed an increasing trend from upstream to downstream. The midstream and downstream had a higher pollutant emission intensity than upstream. At the county scale, most of the counties with high levels of pollution belonged to industrial towns or urbanized regions.

Secondly, the testing results of the OLS showed that there was spatial dependence in the PRXREB, which revealed that environmental pollution had a positive spillover effect on adjacent regions. The increase in pollution degrees in a county would also cause the aggravation of pollution degrees in nearby counties. The effects could be inferred as similar industrial structures and industrial transfer among neighboring counties.

Thirdly, considering the spatial autocorrelation effects of environmental pollution in the PRXREB, econometric models were introduced to analyze the driving factors at the whole scale and local scale. The results showed that the main driving factors in the PRXREB were industrial structure, infrastructure construction, and regulatory measures. Furthermore, since the Guangdong section developed early, these driving factors were mainly revealed in 2012. In contrast, for the Guangxi section, the effects of these factors showed a strengthening trend from 2012 to 2016.

## Figures and Tables

**Figure 1 ijerph-19-06833-f001:**
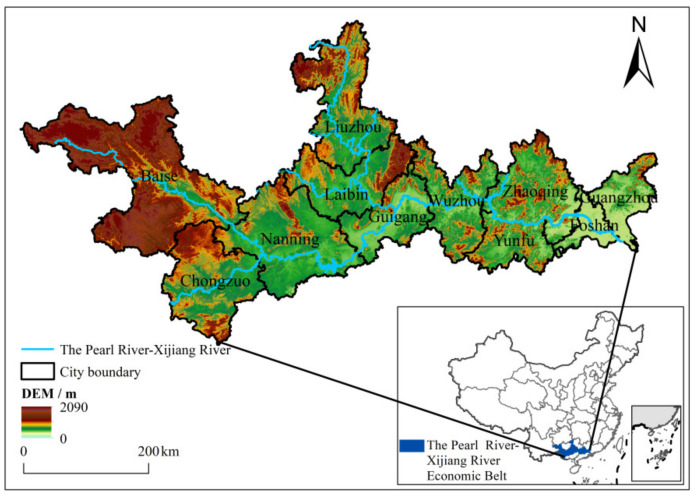
Location of the PRXREB.

**Figure 2 ijerph-19-06833-f002:**
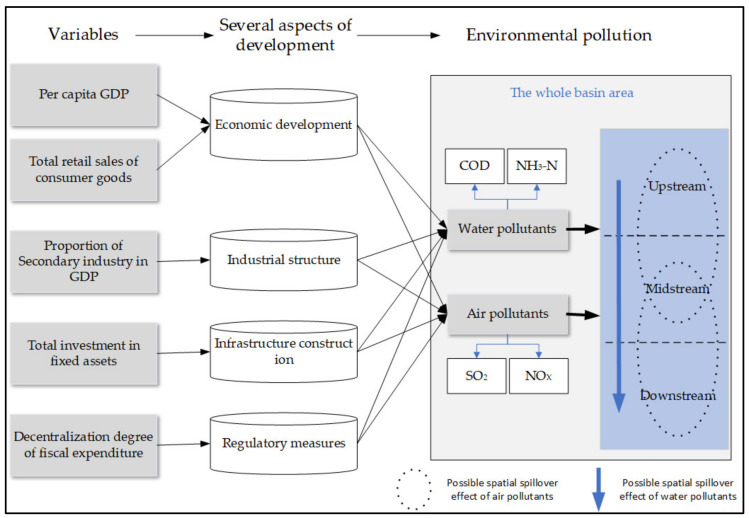
The driving mechanism of environmental pollution in the PRXREB.

**Figure 3 ijerph-19-06833-f003:**
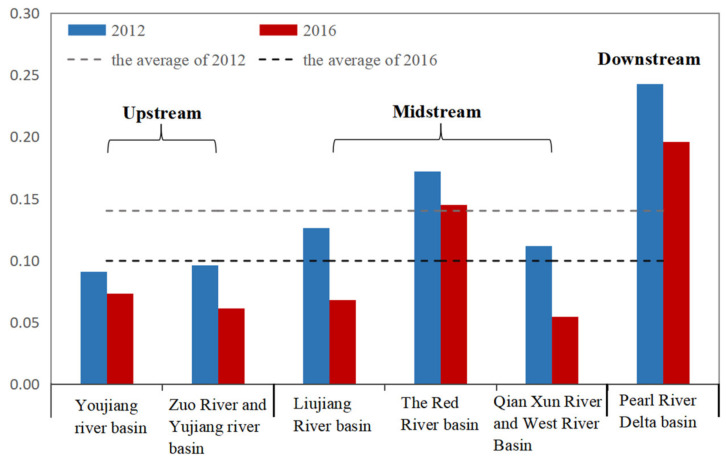
Changes in EPCI in different basins in the PRXREB.

**Figure 4 ijerph-19-06833-f004:**
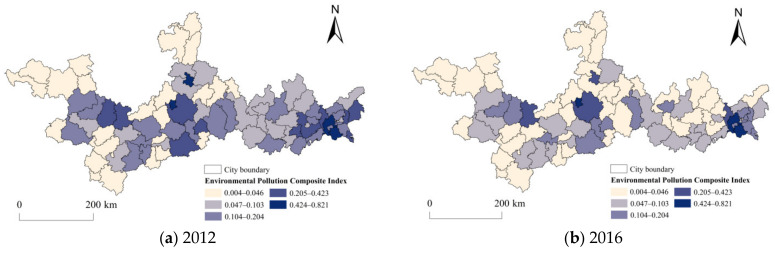
Spatial distribution of EPCI at the county scale in the PRXREB.

**Figure 5 ijerph-19-06833-f005:**
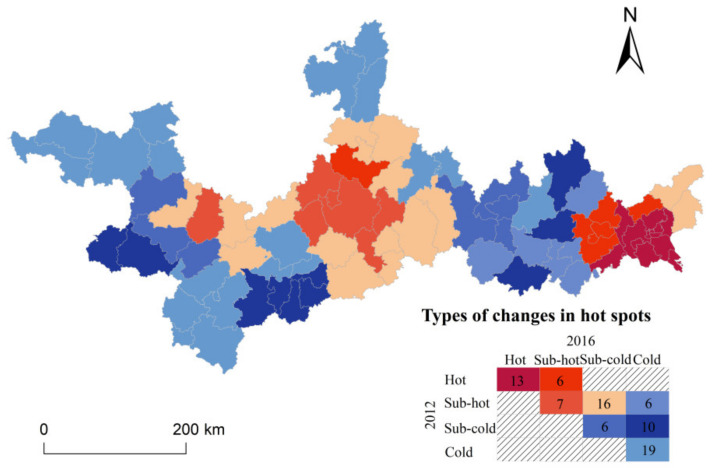
Types of changes in hot spots in the PRXREB.

**Table 1 ijerph-19-06833-t001:** Parameter estimation results of spatial effects in 2012, 2016.

Year	Test	The PRXREB	The Guangxi Section	The Guangdong Section
2012	Moran’s I (error)	3.3944 ***	2.6251 ***	−1.5360 *
	LM-lag	3.1192 *	2.7508 *	2.5600 *
	Robust LM (lag)	0.1825	0.0502	0.0000
	LM-Error	7.4511 ***	4.1499 **	3.9718 **
	Robust LM (error)	4.5144 **	1.4494	1.4118
2016	Moran’s I (error)	2.8526 ***	2.8740 ***	0.0523
	LM-lag	1.2418	3.3421 *	0.3291
	Robust LM (lag)	2.0705	0.0017	0.0035
	LM-Error	2.0705 **	5.5644 **	0.4140
	Robust LM (error)	6.0304 ***	5.5644 *	0.0884

Notes: ***, ** and * indicate significance levels of 0.01, 0.05 and 0.1, respectively.

**Table 2 ijerph-19-06833-t002:** Parameter estimation results of econometric models in 2012 and 2016.

Variables	2012			2016		
	OLS	SEM	SLM	OLS	SEM	SLM
CONSTANT	−12.6668 ***	−12.9131 ***	−11.3242 ***	−6.2156 *	−5.0494 ***	−5.5671 *
lnPGDP	0.0155	0.1216	−0.0301	0.3513 *	0.4223 **	0.3141 *
lnIS	0.9027 ***	0.8587 ***	0.8525 ***	0.8454 ***	0.8063 ***	0.8411 ***
lnSFA	0.4560 ***	0.4582 ***	0.4866 ***	−0.0331 *	−0.1764	−0.1966
lnSC	0.1895	0.1398	0.1356	0.2946 **	0.2322 *	0.2825 **
lnFED	0.2783 *	0.2912 **	0.1355 *	0.5523 ***	0.6919 ***	0.5005 ***
*λ*	-	0.3576 ***	-	-	0.4006 ***	-
*ρ*	-	-	0.2154 *	-	-	0.1559
AIC	190.612	185.124	189.237	226.029	219.42	226.666
SC	205.125	199.637	206.169	240.542	233.933	243.598
R-squared	0.6208	0.6566	0.6401	0.4730	0.5334	0.4846
Log-Likelihood	−89.3059	−86.5618	−87.6186	−107.015	−103.7099	−106.333

Notes: ***, ** and * indicate significance levels of 0.01, 0.05 and 0.1, respectively.

**Table 3 ijerph-19-06833-t003:** Parameter estimation results of econometric models in the Guangxi section.

Variables	2012			2016		
	OLS	SEM	SLM	OLS	SEM	SLM
CONSTANT	−15.4083 ***	−14.164 ***	−12.3648 ***	−5.9894 *	−4.8688	−3.251
lnPGDP	−0.1461	0.0711	−0.0826	0.4957 *	0.5676 **	0.5137 **
lnIS	0.8936 ***	0.8583 ***	0.9152 ***	1.1580 ***	0.9426 ***	1.0953 ***
lnSFA	0.8716 **	0.7250 **	0.7367 **	−0.2851 **	−0.2613 *	−0.2669 *
lnSC	0.0448	0.0374	0.0337	0.4569 **	0.3839 **	0.3830 **
lnFED	0.2322	0.3809	0.3499	0.9538 **	1.0125 ***	1.0572 ***
*λ*		0.4429 ***			0.4369 ***	
*ρ*			0.2808 **			0.333 ***
AIC	133.197	126.744	131.733	151.377	145.893	149.009
SC	145.131	138.678	145.656	163.31	157.827	162.931
R-squared	0.6203	0.6812	0.6512	0.5116	0.5819	0.5628
Log-Likelihood	−60.5986	−57.372	−58.8666	−69.6883	−66.9464	−67.5043

Notes: ***, ** and * indicate significance levels of 0.01, 0.05 and 0.1, respectively.

**Table 4 ijerph-19-06833-t004:** Parameter estimation results of econometric models in the Guangdong section.

Variables	2012			2016		
	OLS	SEM	SLM	OLS	SEM	SLM
CONSTANT	−9.8021 ***	−10.3936 ***	−11.8562 ***	1.8356	1.9802	1.7797
lnPGDP	0.0032	−0.1189	0.0198	0.0858	0.0828	0.0906
lnIS	0.5594 ***	0.5895 ***	0.6512 ***	0.1328	0.1122	0.1331
lnSFA	0.5211 **	0.5754 ***	0.5551 ***	0.1686	0.1846	0.1709
lnSC	−0.0727	−0.0196	−0.0397	−0.1766	−0.1905	−0.1772
lnFED	0.0773	0.0333	0.0619	0.7008 **	0.7099 ***	0.7102 ***
*λ*		−0.6015 ***			−0.079	
*ρ*			−0.2575			−0.0217
AIC	46.3257	42.3899	46.6576	67.7198	67.6443	149.009
SC	54.5295	50.5937	56.2287	75.9236	75.848	162.931
R-squared	0.6043	0.6818	0.6320	0.5016	0.5036	0.5628
Log-Likelihood	−17.1628	−15.1949	−16.3288	−27.8599	−27.8221	−67.5043

Notes: *** and ** indicate significance levels of 0.01 and 0.05, respectively.

## Data Availability

The administrative boundary data were obtained from the website of the National Basic Geographic Information System, and the pollutant discharge and socio-economic data were obtained from China Environmental Statistical Yearbook, China Environmental Yearbook, China City Statistical Yearbook, China Regional Economic Statistical Yearbook, and provincial statistical yearbook.

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
