# Peer review of "Study on Driving Factors and Spatial Effects of Environmental Pollution in the Pearl River-Xijiang River Economic Belt, China"

_ijerph, 2022, doi:10.3390/ijerph19116833_

Round 1
Reviewer 1 Report
the manuscript entitled "The Study on Driving Factors and Spatial Effects of Environmental Pollution in the Pearl River-West River Economic Belt, China" has a good work of study. The strength of the study is on their model and technical insights of conducting the study. However, the main reason for not recommending it is because there's a lot of flaws on the write up. It is using the entropy weight method for EPCI and spatial error model for the driving force and spatial effects.
I do not recommend this manuscript for publication in this form. This manuscript has many more issues and concerns. The writing of the statement in the manuscript is not pretty good. It is necessary to adjust the structuring of words and adjust the grammar as well. Suggestions and comments are the following:
- Page 14 line 384-435: Formatting of Mathematical models should be presented in your introduction or if necessary in the methodology. These statements should be presented in the introduction or if necessary in the methodology. These statements (pages 14-15 linse 384-435) are the same statements in the introduction (pages 3-4, lines 115-168).
- There's a redundant statement in the Data and Methods Section of the manuscript (page 3 - 4, line 115 - 168).
- Another thing that is noticeable here is the statement in page 17 lines 507-513. The statement is too long and suggests that this needs revisions.
- Another noticeable statement is on page 2 lines 95-97 that needs revised in layman's terms to make it more understandable for the readers.
Reviewer 2 Report
Review ijerph-1673550
Study on Driving Factors and Spatial Effects of Environmental Pollution in the Pearl River-West River Economic Belt, China
by Yutian Liang, Jiaxi Zhang and Kan Zhou
The work uses advanced methods of spatial analysis and correlation between many factors that can be expressed numerically. The equation that describes these relationships is a linear equation. Unfortunately, the data taken into account concern only two points on the time scale, so by assumption some coefficients of the model equation can be calculated and obtain a high correlation. However, the authors with different models obtained R squared in the range 0.47 to 0.68, which is a weak to moderate correlation. Conclusions based on such correlations may not be well founded.
In my opinion, the authors should take into account a larger scope of time maybe analyze the data also for 2014, 2018 and 2020.
The second problem in the assumptions of this analysis is that the dependence of the pollutant emission spatial distribution is not a natural law - just like the laws of physics. On the other hand, the course of phenomena on a statistical scale is influenced by administrative decisions and a change in the law.
A few specific comments.
Line 33 It is not known what the authors mean when they write about "environmental capacity".
Lines 40-97 are one paragraph. They should be divided into several paragraphs so that the reader can get an idea of the main theses of the paper.
Chapter 2 Data and methods should have clearly marked sub-titles:
2.1 Case overview
2.2 Data
2.3 Environmental pollution composite index
2.4 Spatial economic model
2.5 Set Y as the explained variable
Lines 170-172 Looks like the Editor's comments to the author of the scientific article. In fact, this is an excerpt from the instructions for the authors: https://www.mdpi.com › files › molecules-template
Lines 201 "According to Jenks ..." should be explained, so that not mathematicians could understand the essence of this approach. One could write "According to Jenks natural breaks optimization ..."
Lines 221-227 The concepts of "Morans' I" should be explained to a reader unfamiliar with these methods. Similarly, the meaning of “Hot”, “Sub-hot”, “Sub-cold” and “Cold” should be described what in practice “Hot” means. Does it mean high values or a high spread of values?
Lines 239-249 contain the obvious statements resulting from common knowledge and common sense. However, they do not result from previous calculations.
Line 263-264 The sentence is trivial, because in such a large area there must be a variety of land, population and industry. Cities and then industry arise where there is access to water and other resources.
Lines 368-383 are a repetition of the tables and figures shown earlier
Lines 384-435 They are the exact repetition of the text from lines 115-168, except for the difference in the numbering of the references in lines 151 and 419
Lines 448-454 The sentence is not grammatical and should be divided into several shorter ones.
Conclusions contain recommendations for the authorities of the analysed region. However, in view of the rapid industrial and population development in this area, detailed recommendations are much late because they are based on data from 2016. The analysis is very late - based on 2016 data and published in 2022. During this time, a lot has changed in China, possibly including environmental standards and then conclusions are questionable..
Round 2
Reviewer 1 Report
The manuscript is now acceptable in thins form. There are only very very few and minor issues in using an appropriate words in the manuscript . I suggests that the authors may review it once more for final proof reading.
